# Investigation of the Microstructure of Fine-Grained YPO$_4$:Gd Ceramics with Xenotime Structure after Xe Irradiation

Dmitriy A. Mikhaylov [1,*], Ekaterina A. Potanina [1], Aleksey V. Nokhrin [1,*], Albina I. Orlova [1], Pavel A. Yunin [1,2], Nikita V. Sakharov [1], Maksim S. Boldin [1], Oleg A. Belkin [1], Vladimir A. Skuratov [3,4,5], Askar T. Issatov [3,6,7], Vladimir N. Chuvil'deev [1] and Nataliya Y. Tabachkova [8,9]

1   Materials Science Department, Physical and Technical Research Institute, Lobachevsky State University of Nizhny Novgorod, 603022 Nizhny Novgorod, Russia; potanina@nifti.unn.ru (E.A.P.); albina.orlova@gmail.com (A.I.O.); yunin@ipmras.ru (P.A.Y.); nvsaharov@nifti.unn.ru (N.V.S.); boldin@nifti.unn.ru (M.S.B.); nikleb64@mail.ru (O.A.B.); chuvildeev@nifti.unn.ru (V.N.C.)
2   Laboratory of Diagnostics of Radiation Defects in Solid State Nanostructure, Institute for Physics of Microstructure, Russian Academy of Science, 603950 Nizhniy Novgorod, Russia
3   G.N. Flerov Laboratory of Nuclear Reactions, Joint Institute of Nuclear Research, 141980 Dubna, Russia; skuratov@jinr.ru (V.A.S.); issatov@jinr.ru (A.T.I.)
4   Institute of Nuclear Physics and Engineering, National Research Nuclear University MEPhI (Moscow Engineering Physics Institute), 115409 Moscow, Russia
5   Department of Nuclear Physics, Dubna State University, 181982 Dubna, Russia
6   International Department of Nuclear Physics, New Materials and Technologies, The Faculty of Physics and Technology, Gumilov Eurasian National University, Nur-Sultan 010000, Kazakhstan
7   Laboratory of Nuclear Processes, Nuclear Physics Department, The Institute of Nuclear Physics, Almaty 050032, Kazakhstan
8   Center Collective Use "Materials Science and Metallurgy", National University of Science and Technology "MISIS", 119991 Moscow, Russia; ntabachkova@misis.ru
9   Laboratory "FIANIT", Laser Materials and Technology Research Center, A.M. Prokhorov General Physics Institute, Russian Academy of Sciences, 119991 Moscow, Russia
*   Correspondence: mikhaylov@ichem.unn.ru (D.A.M.); nokhrin@nifti.unn.ru (A.V.N.)

**Abstract:** This paper reports on the preparation of xenotime-structured ceramics using the Spark Plasma Sintering (SPS) method. Y$_{0.95}$Gd$_{0.05}$PO$_4$ (YPO$_4$:Gd) phosphates were obtained using the sol-gel method. The synthesized powders were nanodispersed and were agglomerated (the agglomerates sizes were 10–50 µm). The ceramics had a fine-grained microstructure and a high relative density (98.67 ± 0.18%). The total time of the SPS process was approximately 18 min. The sintered high-density YPO$_4$:Gd ceramics with a xenotime structure were irradiated with $^{132}$Xe$^{+26}$ ions with 167 MeV of energy and fluences in the range of $1 \times 10^{12}$–$3 \times 10^{13}$ cm$^{-2}$. Complete amorphization was not achieved even at the maximum fluence. The calculated value of the critical fluence was $(9.2 \pm 0.1) \times 10^{14}$ cm$^{-2}$. According to the results of grazing incidence X-ray diffraction (GIXRD), the volume fraction of the amorphous structure increased from 20 to 70% with increasing fluence from $1 \times 10^{12}$ up to $3 \times 10^{13}$ cm$^{-2}$. The intensity of the 200 YPO$_4$:Gd XRD peak reached ~80% of the initial intensity after recovery annealing (700 °C, 18 h).

**Keywords:** xenotime; ceramics; spark plasma sintering; microstructure; GIXRD; metamict phase

## 1. Introduction

Compounds with xenotime structures are among the potential matrices for actinide immobilization [1–3]. The xenotime structure consists of PO$_4$ tetrahedrons and YO$_8$ polyhedrons, and it crystallizes in the tetragonal syngony (space grout $I4_1/amd$) [4]. The structure is characterized by a wide set of isomorphic forms and may incorporate numerous elements, including lanthanides (from Tb to Lu) and rare earth (Y, Sc [4], Pu, Cm, Np [5–7], Gd, Dy, Er, Yb [8,9], Th, U [2,9,10]). Due to the presence of Th and U in the composition, natural xenotime compounds can be exposed to radiation with a fluence of up

to $(1.4$–$14) \times 10^{16}$ $\alpha$/mg [10]. This characterizes the xenotime structure as having a good resistance to self-irradiation. In addition, compounds with a xenotime structure have a high melting point [11,12] and are stable under hydrolytic conditions [11,13].

The amorphization of synthetic xenotime samples under irradiation from $Kr^{2+}$ ions (800 keV) was investigated as a function of temperature (20–600 K) in [14]. Xenotime was shown to be more susceptible to radiation at elevated temperatures. Synthetic $LuPO_4$ containing 1.0 wt.% $^{244}$Cm accumulated a fluence of $5 \times 10^{16}$ $\alpha$/mg after 18 years of exposure and remained in a highly crystalline state. However, some defect regions were found in the samples, which could contain accumulated radiogenic helium [10]. The implantation of Au ions (2 MeV, $1 \times 10^{14}$, $5 \times 10^{14}$, and $1 \times 10^{15}$ $cm^{-2}$) caused structural damage in phosphates, $La_{1-x}Yb_xPO_4$ (x = 0.7, 1.0) [15]. After irradiation with maximum fluence, the samples recrystallized partially. After annealing at 300 °C, they recovered completely. In [16], $YPO_4$ ceramics were implanted with $Au^-$ ions with different energies (35, 22, 14, and 7 MeV) and fluences ($1.6 \times 10^{13}$–$6.5 \times 10^{13}$ $cm^{-2}$). The authors supposed that the heterogenous damaged layer observed in the $YPO_4$ ceramics was a result of epitaxial annealing occurring at the crystal-amorphous phase interface. The irradiation of $ErPO_4$ ceramics with Au and He ions was studied in [17]. He ions were found to partially prevent the amorphization of the sample during co-irradiation.

Ceramics based on compounds with xenotime structures are usually obtained by pre-pressing powders followed by conventional sintering at temperatures of ~1300–1600 °C; the sintering times are 1–5 h. This allows for the attainment of ceramics with a relative density of ~90% [18,19]. To ensure an increased radiation and hydrolytic resistance in xenotime-structured ceramics, it is necessary to provide higher density. In [20], a $YPO_4$ ceramic with a relative density of 98% was obtained with a long sintering at 1600 °C during 10 h. The authors of [21] suggested that the sintering of $YPO_4$ ceramics is difficult, probably owing to features of their particle morphology—anisotropic crystal growth and the formation of needle-shaped particles takes place when they are heated up to 1400 °C.

Spark plasma sintering (SPS) is a promising method of obtaining ceramics for the immobilization of high-level waste (HLW) [22–26] and obtaining materials for the nuclear industry [27,28]. SPS is a new method of hot pressing featuring high heating rates (up to 2500 °C/min) [22,26]. The possibility of achieving an increased density in ceramics in short sintering times is an important advantage of SPS technology [24–26,29,30]. This plays an important role in the handling of HLW. Mineral-like ceramics obtained with SPS have a high relative density, are hydrolytic, and are radiation resistant [25,29–39]. The compaction kinetics of the nano- and submicron particles in SPS are determined by the intensity of the grain boundary diffusion [35–42]. The grain boundary diffusion coefficient for fine-grained ceramics at low heating temperatures is known to be several orders of magnitude greater than the crystal lattice diffusion coefficient [43,44]. This allows for the attainment of ceramic specimens with a high relative density by using SPS at low temperatures and increased heating rates [25,29–37].

The goal of the present work was to obtain $Y_{0.95}Gd_{0.05}PO_4$ ($YPO_4$:Gd) high-density ceramics with xenotime structures by using SPS and to study their microstructures and properties. In particular, we studied the effect of irradiation on the microstructure and properties of $YPO_4$:Gd ceramics, including the formation of the amorphous phase. $Gd^{3+}$ was chosen as a simulator of $Cm^{3+}$ due to the similarity of their electronic configurations and ionic radii and, therefore, their chemical and physical properties. Irradiation with heavy ions in a charged particle accelerator simulated harsh radiation conditions, allowing for the express assessment of the radiation resistance of the materials.

## 2. Materials and Methods

The $Y_{0.95}Gd_{0.05}PO_4$ phosphate powders were obtained with the sol-gel method. Crystalline $Y(NO_3)_3 \cdot 6H_2O$ and $Gd_2O_3$ dissolved in the excess of a weakly acidic solution of $HNO_3$ (pH = 4–5) were used as the initial reagents. Ammonium dihydrogen phosphate $NH_4H_2PO_4$ was used as a precipitant. A 1 M solution of $NH_4H_2PO_4$ was added dropwise

with continuous stirring. The resulting gel was vigorously stirred further for 5 min until complete homogenization was achieved and dried at 90 °C for a day. The dry residue was dispersed in an agate mortar. The resulting solid mixture was heated step-by-step to 600, 700, 800, and 900 °C, holding for 5 h at each stage without dispersion between the heating stages.

$YPO_4$:Gd ceramic samples of 10 mm in diameter and 3 mm in height were obtained with the SPS method using Dr. Sinter® model SPS-625 equipment (NJS Co., Ltd., Tokyo, Japan). Sintering was carried out in a vacuum (5–6 Pa) in graphite molds. To improve the contact of the powder sample with the graphite mold and to compensate the difference in the thermal expansion coefficients of graphite and the ceramics, a graphite foil was placed inside the mold. A uniaxial pressure P = 70 MPa was applied simultaneously at the start of heating. Two-stage heating was used: Stage I—heating up to 600 °C with a heating rate of $V_h$ = 100 °C/min; Stage II—heating with $V_h$ = 50 °C/min up to the sintering temperature, $T_s$ (Figure 1a). The holding time at the sintering temperature, $T_s$, was $t_s$ = 2 min. Total time of the SPS process was approximately 18 min. The sample was cooled down together with the Dr. Sinter® model SPS-625 setup.

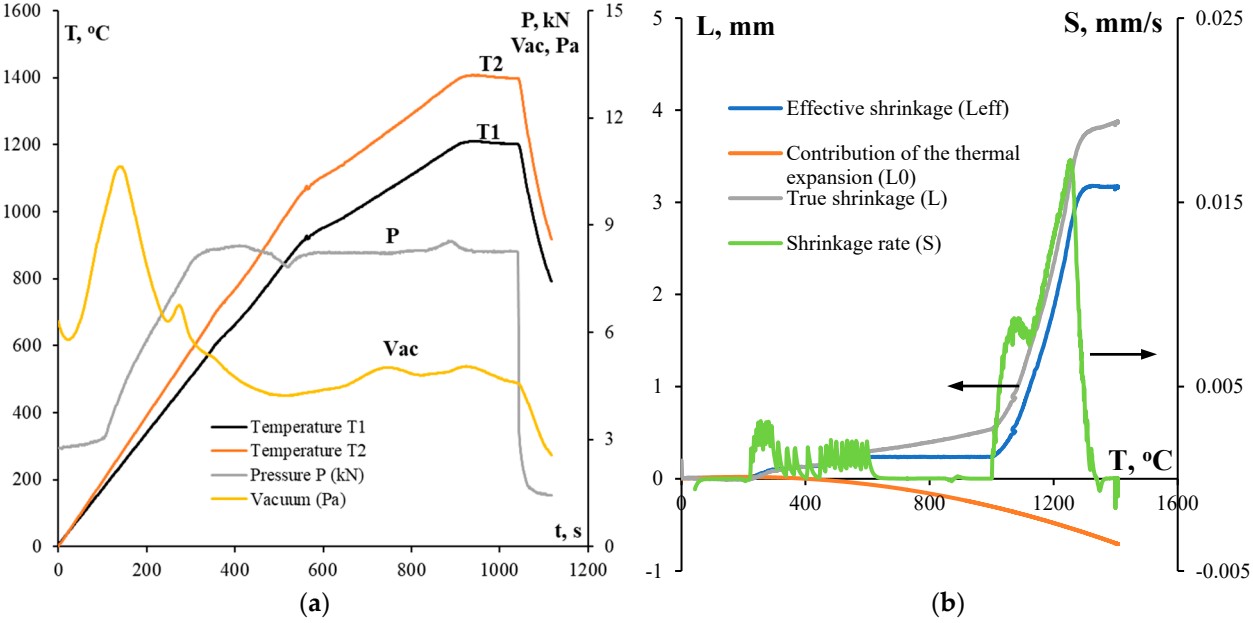

**Figure 1.** SPS diagrams for $YPO_4$:Gd ceramics: (**a**) dependences of temperature (T), applied pressure (P), and vacuum pressure (Vac) on the time of the SPS process; (**b**) dependence of the shrinkage on the heating temperature.

The temperature of the sample was measured using CHINO® IR-AHS2 infrared pyrometer (Chino Corporation, Tokyo, Japan) focused on the surface of the graphite mold. Based on previous studies, a comparison of the data measured with the optical pyrometer (curve T1 in Figure 1a), and an additional thermocouple attached to the sample surface, the values of T1 were recalculated to the actual sample temperature (T2) using the following empirical equation: T2 = 1.1686·T1 − 43.416 (Figure 1a).

The dependencies of the effective shrinkage ($L_{eff}$) in the SPS process on the sintering time and the heating temperature were recorded. To take into account the contribution of thermal expansion ($L_0$), special experiments were carried out by heating the empty molds (see [45]). The true shrinkage value was calculated as $L = L_{eff} - L_0$ (Figure 1b). Using the temperature curve L(T) in the linear approximation, the temperature dependence of the shrinkage rate was calculated: $S = \Delta L / \Delta t$.

After sintering, the surfaces of the sample were contaminated with residual graphite paper. To remove the residual graphite, the samples were annealed in EKPS-10 air furnace

(Smolensk SKTB SPU, JSC., Smolensk, Russia) at 750 °C for 1 h. The samples were heated up and cooled down together using the furnace.

X-ray diffraction (XRD) phase analysis of the powders and ceramics was carried out using Shimadzu® LabX™ XRD-6100 diffractometer (Shimadzu Co., Kyoto, Japan). The semiquantitative phase analysis was performed using PhasanX® v.2.0 software.

The grazing incidence XRD (GIXRD) phase analysis of the irradiated ceramics was performed using a Bruker® D8 Discover™ X-ray diffractometer (Bruker Co., Billerica, MA, USA). An X-ray tube with a Cu cathode ($CuK_\alpha$ radiation) was used. The investigations were carried out in the parallel beam geometry with a parabolic Göbel mirror, a round collimator of 1 mm in diameter on the primary and a 0.2° Soller slit in front of the detector. In every series of GIXRD experiments, the angle of incidence of the primary beam on the specimen ($\alpha$) scanned from 2° to 10°. In each experiment, the XRD curves were recorded by scanning the detector in angle 2θ with the range θ = 25°–28° corresponding to the position of the 200 XRD peak of $YPO_4$.

The microstructure of the powders and ceramics was investigated using Jeol® JSM-6490 Scanning Electron Microscope (SEM, Jeol® Ltd., Tokyo, Japan) with Oxford Instruments® INCA 350 Energy Dispersion Spectroscopy (EDS) microanalyzer (Oxford Instruments® pls., Abingdon, UK) and Jeol® JEM-2100 Transmission Electron Microscope (TEM, Jeol® Ltd., Tokyo, Japan). The mean particle size (R) and the mean grain size (d) was calculated with the section method using GoodGrains® 2.0 software (UNN, Nizhny Novgorod, Russia). The microstructure of the ceramic surface layers after irradiation was studied using a Leica® IM DRM metallographic optical microscope (Leica Microsystems, Wetzlar, Germany).

The density of the sintered ceramics was measured with hydrostatic weighing in distilled water using Sartorius® CPA 225D balance (Sartorius AG, Göttingen, Germany). The uncertainty of the density measurement was ±0.001 $g/cm^3$. The theoretical density of the ceramics ($\rho_{th}$) was calculated on the basis of the XRD investigations.

The radiation resistance of the ceramics was evaluated using high-energy (167 MeV) $Xe^{+26}$ ion irradiation in an IC-100 FLNR JINR cyclotron (Joint Institute of Nuclear Research, Dubna, Russia). The samples were irradiated at temperatures of 23–27 °C with fluences (F) of $1 \times 10^{12}$–$3 \times 10^{13}$ $cm^{-2}$. The average ion flux was limited to ≈2 × $10^9$ $cm^{-2}$ $s^{-1}$ to avoid significantly heating the targets. The temperature of the targets during irradiation did not exceed 30 °C. The uniform distribution of the ion flux over the irradiated target surface was achieved by ion beam scanning. The accuracy of the ion flux and fluence measurements was 15%.

## 3. Results and Discussion

According to the XRD data (curve 1 in Figure 2), the synthesized $Y_{0.95}Gd_{0.05}PO_4$ phosphate powder was monophasic and was identical to the analogue (ICDD PDF 83-0658, sp.gr. $I4_1/amd$, $a = b = 6.8905 \pm 0.0003$ Å, $c = 6.0227 \pm 0.0004$ Å, $\alpha = \beta = \gamma = 90°$). The theoretical density of the $YPO_4$:Gd compound, calculated from the analysis of the results of the XRD studies, is $\rho_{th} = 4.349 \pm 0.001$ $g/cm^3$. According to the XRD phase analysis results, there were no impurity phases in the powders.

The SEM results show the synthesized powders form large agglomerates with sizes ranging from 10 to 50 μm (Figure 3a). The agglomerates consist of nanoparticles packed closely to each other (Figure 3b). The conclusion that the agglomerates consist of individual nanoparticles was confirmed by the results of TEM studies (Figure 4). No large, needle-shaped particles were observed in the powders (see [21]). This is probably related to the synthesis of the powders at lower temperatures in the present work than in [21].

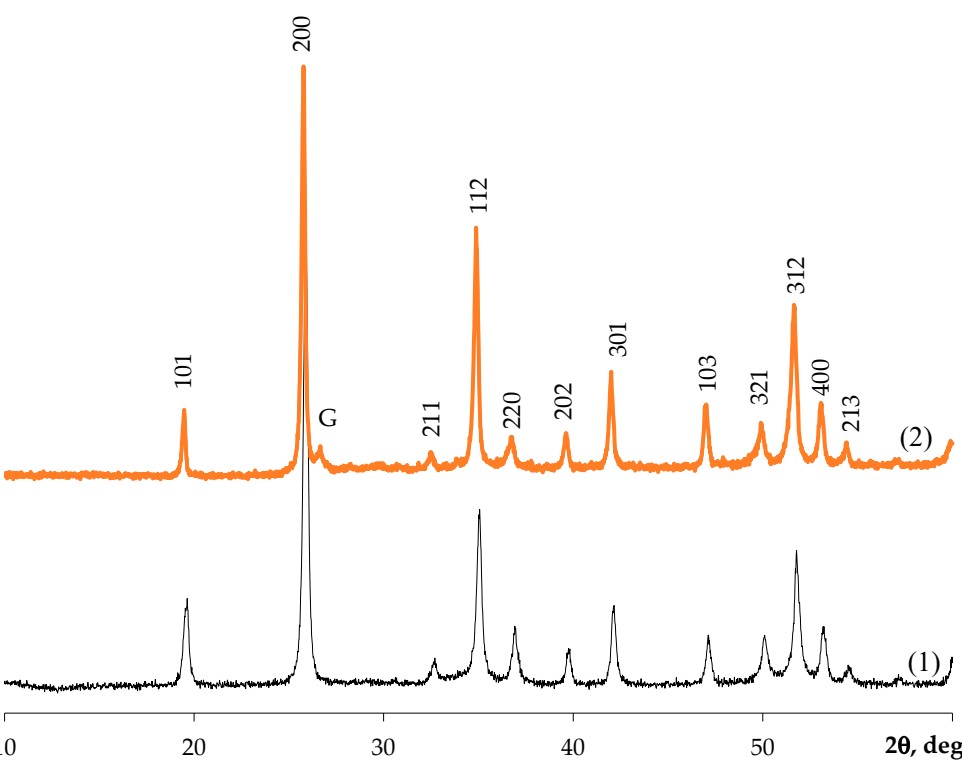

**Figure 2.** XRD data for YPO$_4$:Gd powder after annealing at 900 °C (5 h) (1) and for the SPS ceramic (2). Symbol "G" marks the reflection from graphite.

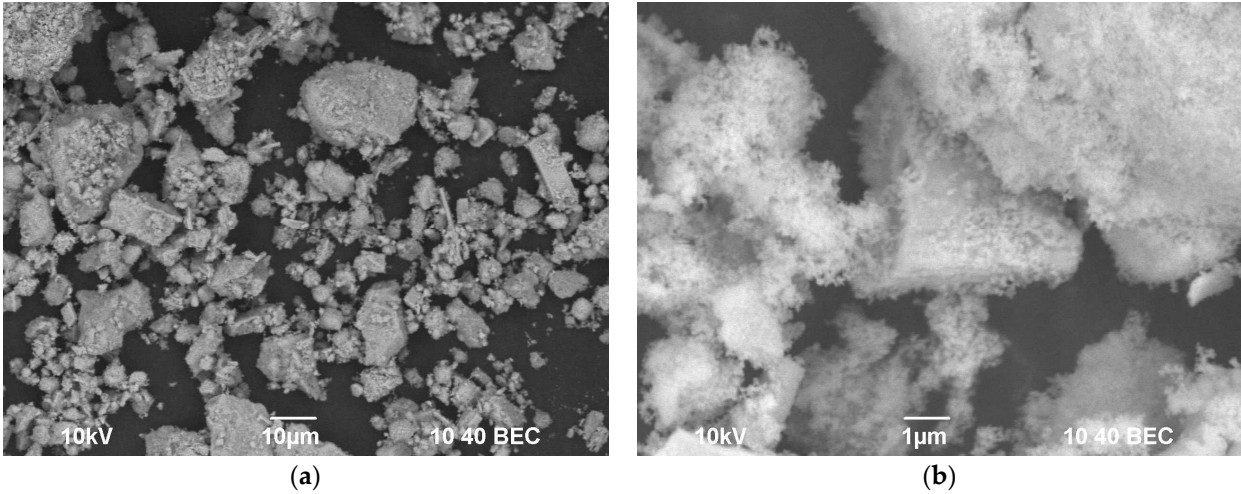

(**a**)　　　　　　　　　　　(**b**)

**Figure 3.** SEM images of YPO$_4$:Gd powder (**a**,**b**). (**a**,**b**) present different images of the same powder at different magnifications.

One can see in Figure 4a that the YPO$_4$:Gd powder consisted of agglomerates, which, in turn, consisted of nanoparticles. The sizes of the agglomerates varied in a broad range —from several microns to several tens of microns. Closed pores were observed in some particles. The particles had a crystalline structure. This was confirmed with diffraction patterns measured from a selected area and the high resolution image of the particles (Figure 4c,d). The positions of the XRD peaks in the electron diffraction pattern (Figure 4b) point to a YPO$_4$ phase. The broadening of the rings in the electron diffraction pattern also evidences the fine and dispersed microstructures of the powder. The large particles in the powders were mainly elongated. The shapes of smaller particles were close to the spherical ones. In the statistical analysis of the size distribution of the particles, the shapes of the

elongated particles were approximated by ellipses. When plotting the size distributions of the particles (Figure 4c), both the bigger and smaller axes of the ellipses were taken into account. The particle sizes ranged from 20 to 90 nm. The maximum of the particle size distribution corresponds to 50 nm (Figure 4c).

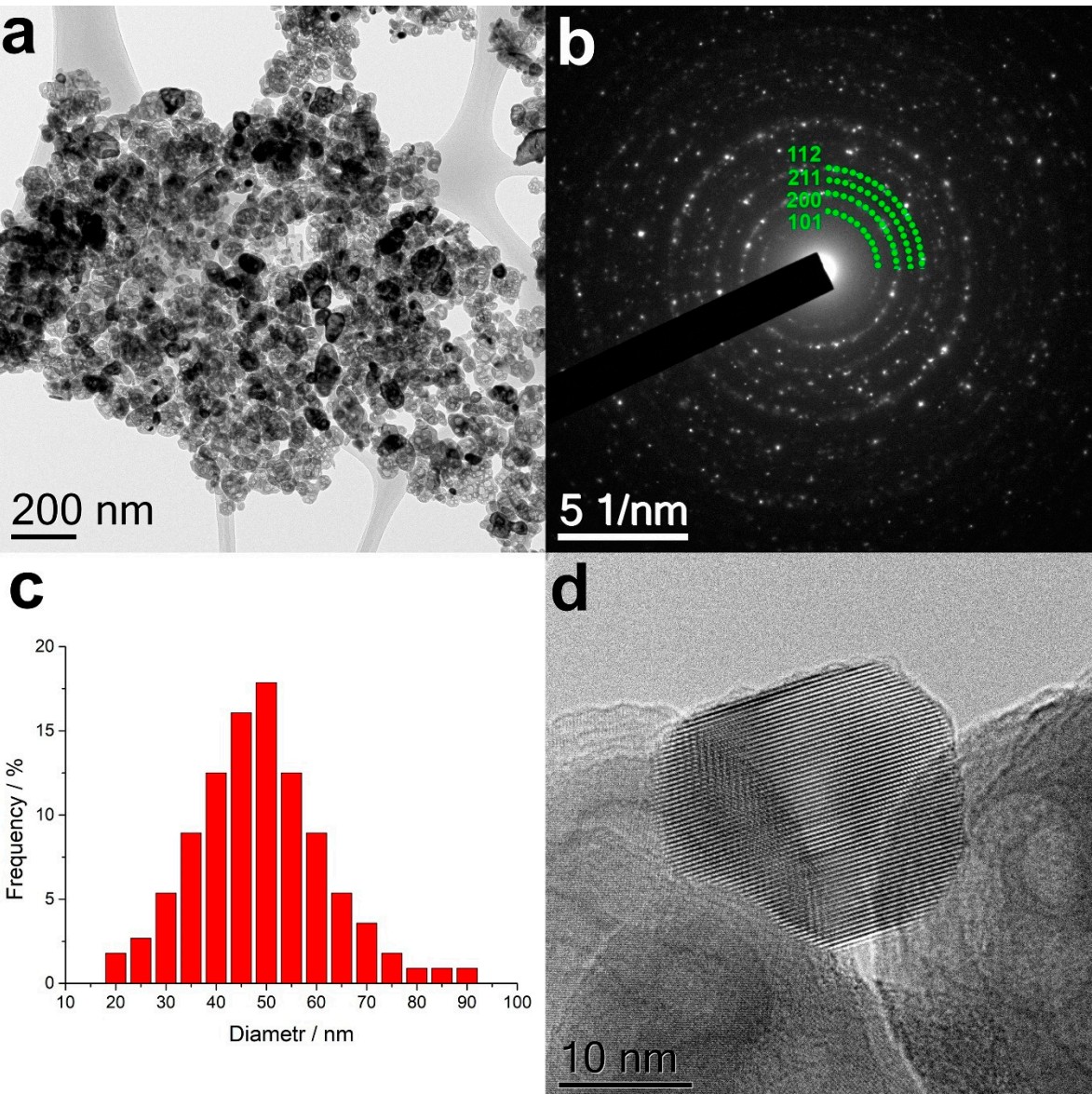

**Figure 4.** Results of TEM studies of $YPO_4$:Gd phosphate powder: (**a**) bright field image of the powder, (**b**) electron diffraction pattern, (**c**) particle size distribution histogram, and (**d**) the crystal structure of the particles.

Seven identical ceramic samples were fabricated with SPS. The relative density of the sintered ceramic samples was $98.67 \pm 0.18\%$. The ceramic samples had no external damage or macrodefects.

Figure 1a shows the sintering diagrams "time t—temperature T—pressure P—vacuum pressure Vac" for the $YPO_4$:Gd ceramics. Note that the vacuum pressure during heating remains approximately constant (4.5–4.8 Pa). This indicates the absence of decomposition in the $YPO_4$:Gd phase during SPS as well as the absence of significant dissociation in elements from the ceramic sample surfaces. Figure 1b presents the temperature dependencies of the shrinkage (L) and of the shrinkage rate (S) for $YPO_4$:Gd powders. The curve, L(T), has the usual three-stage character—Stage I, with a low compaction intensity (T < 1000 °C),

Stage II, with an intensive compaction of powders in the temperature range from 1000 to 1300 °C, and, finally, Stage III, where the intensity of powder shrinkage becomes small again (T > 1300 °C).

It is interesting to note that two maxima are clearly visible in the temperature dependence of the shrinkage rate: at ~1100 °C (the maximum shrinkage rate $S_{max}$ reached the value of ~$8 \times 10^{-3}$ mm/s) and another one at ~1250 °C ($S_{max} \sim 17 \times 10^{-3}$ mm/s). In our opinion, the two-stage character of the curve, S(T), was related to the microstructure of the YPO$_4$:Gd powders (Figures 3 and 4). At the first stage, the YPO$_4$:Gd phosphate nanoparticles are sintered inside the agglomerates. At the second stage, the agglomerates are sintered to each other.

It is important to note that the sintering temperature for the ceramics sintered from the powder with the xenotime structure by using SPS was lower than the one in other processes reported in the literature [18–20,25]. Furthermore, the duration of the SPS process was much shorter as compared to other methods. The results obtained indicate the high workability and efficiency of SPS for producing ceramics from polycrystalline inorganic powders, including the YPO$_4$:Gd compound with the xenotime structure studied in the present work. A significant decrease in the temperature and duration of the sintering process in SPS makes it possible to prevent the growth of crystallites and to obtain a material with a high relative density, low porosity, and denser microstructure.

According to the XRD results, the phase composition of the ceramic samples was similar to that of the initial powders (curve 2 in Figure 2). There was a low intensive graphite peak in the XRD pattern of the ceramics. The presence of this peak was associated with sintering in the graphite mold.

SEM images of the fractures in the ceramic samples are shown in Figure 5. The micrographs show the ceramic to have a highly dense, fine-grained microstructure. The majority of the ceramic grains were from 5 to 15 μm in size (Figure 5a). Furthermore, there were some parts with a fine-grained microstructure where the grain sizes were 1–2 μm. In addition, there were some isolated pores within these fine-grained microstructures. In some parts of the samples, large grains were present. In our opinion, this is a consequence of the presence of large particles and agglomerates in the original powders, not of the accelerated grain growth during sintering with SPS.

The results of the metallographic studies presented in Figure 5 also indicate the formation of a highly dense, fine-grained microstructure with a grain size of ~15–30 μm. Thus, rather intensive grain growth took place during SPS. It is interesting to note that the grain sizes in the ceramics (~10–20 μm, see Figure 5a) were close to the sizes of the agglomerates in the initial powder (see Figure 3a). The fractographic analysis of the fractures revealed parts consisting of submicron particles in some samples. The sizes of these parts were also close to the sizes of the agglomerates and to those of the grains (Figures 3a and 5b). In our opinion, the results obtained also indirectly indicate that the sintering process has a two-stage character (see above). There were some large, elongated grains in the central parts of the samples (Figure 5c,d). Note that the SPS temperature for the YPO$_4$:Gd ceramics in the present work corresponded to the synthesis temperature for the YPO$_4$ powders in [21], where anisotropic particle growth was reported. This suggests that the large elongated grains in the YPO$_4$:Gd ceramics made with SPS also originate from anisotropic particle growth along selected crystallographic direction.

The ceramic samples were irradiated with Xe$^{+26}$ ions with fluences of F = $1 \times 10^{12}$, $3 \times 10^{12}$, $7 \times 10^{12}$, $1 \times 10^{13}$, and $3 \times 10^{13}$ cm$^{-2}$. The XRD results for the irradiated samples are presented in Figure 6a. One can see in Figure 6a,b that increasing the fluence leads to a decrease in the relative intensity of the 200 YPO$_4$:Gd XRD peak. It is interesting to note that the XRD peak from graphite became much more visible in the irradiated ceramics than in the nonirradiated ones.

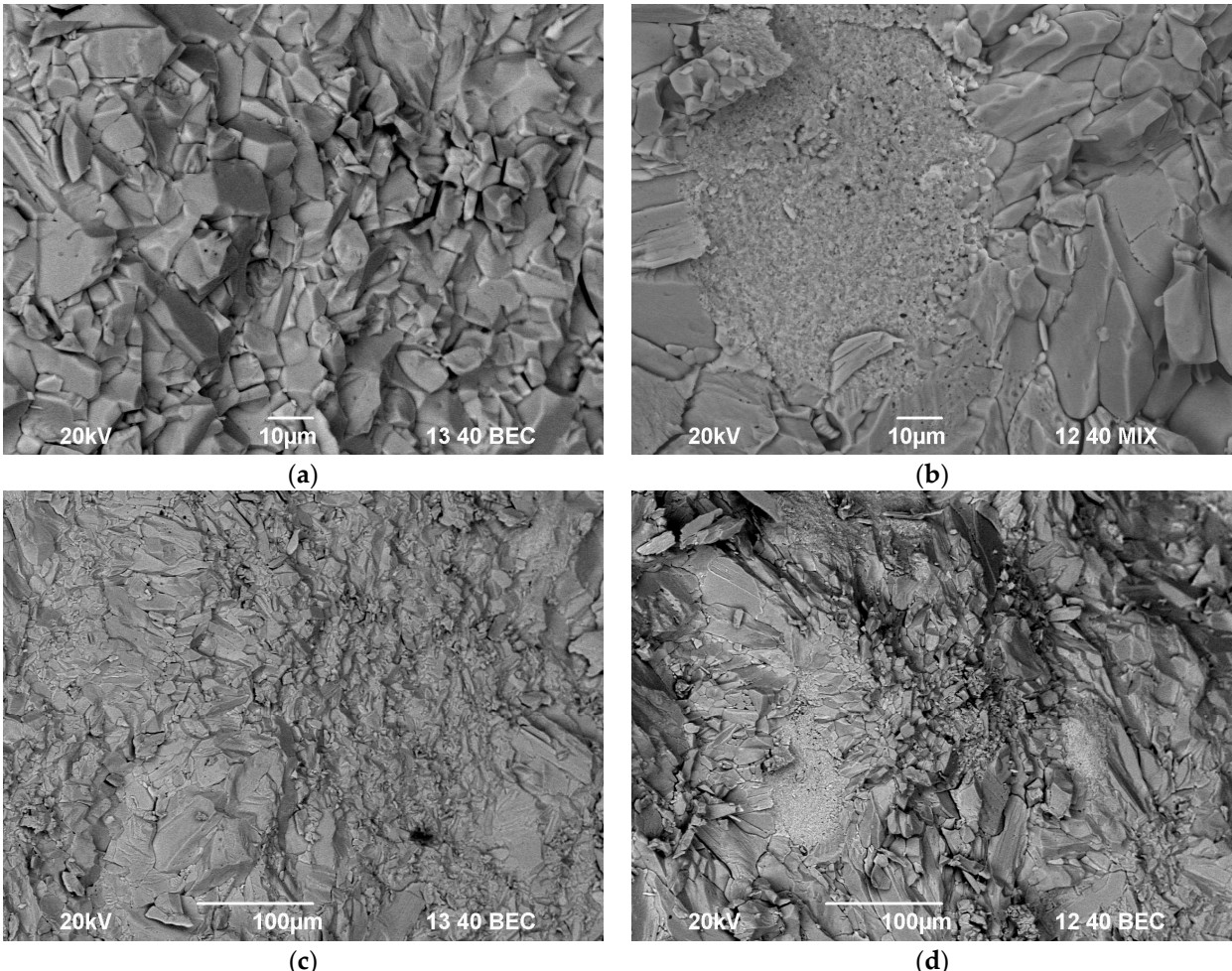

**Figure 5.** Microstructure of the $YPO_4$:Gd ceramics obtained with SPS. SEM of the samples fracture surface.

The diffraction pattern observed indicates the absence of a complete amorphization of the surface layer of the ceramic sample: the XRD peaks corresponding to the $YPO_4$:Gd phase can be seen quite clearly even after irradiation with a fluence of $3 \times 10^{13}$ cm$^{-2}$ (Figure 6a). We emphasize that the broadening and decrease in the intensity of the XRD peaks (at the diffraction angles $2\theta = 25$–$28°$) indicate an increased fraction of the amorphous phase on the sample surfaces. The dependencies of the 200 XRD peak intensity on the fluence are shown in Figure 6b,c. As one can see in Figure 6c, an exponential decay in the 200 XRD peak intensity with an increasing fluence was observed.

A metallographic analysis of the side surface of the irradiated ceramic samples revealed the 10–20 μm-thick surface layers had another color (Figure 7).

In our opinion, the change in the surface layer color originated from the carbonization of the ceramics owing to the interaction of the graphite mold and the graphite foil with the ceramic sample surface. It is interesting to note that the carbonized layers were manifested more clearly in the irradiated samples than in the initial ones. Note also that the carbonization of the surface layers in SPS has been observed often for various metallic alloys [26,46,47] and ceramic materials [26,46,48–55]. Some authors suggest the carbonization of materials in SPS originates from the low-temperature decomposition of the polymer used to bind the graphite foil [54]. The authors of [54] suggested that gas-forming CO released from the graphite foil while heating was the source of the carbon "contamination" in the ceramic surface. In the course of rapid heating, the gas-phase CO penetrates the inside of the specimens through open pores and then appears to be "locked" inside the pores upon achieving a high relative density. To suppress the intensive diffusion of carbon

inside ceramic surfaces, sintering regimes with increased pressure [55] with a step-wise change in the temperature and heating rate of SPS [55], the application of molybdenum foil to protect the ceramics [49], etc. were proposed.

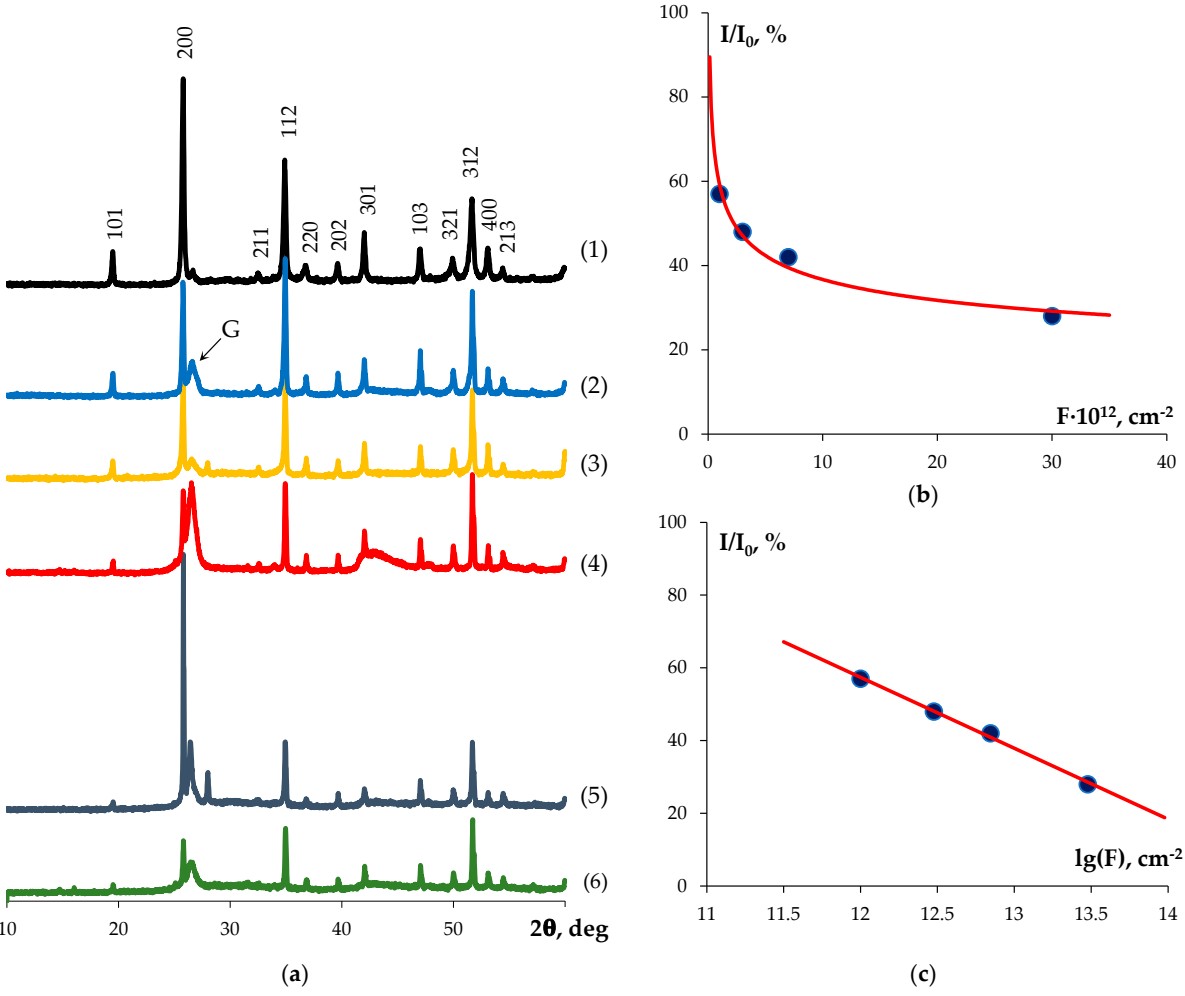

**Figure 6.** XRD data for the $YPO_4$:Gd ceramics after irradiation with $Xe^{+26}$ ions (**a**). Fluences, $cm^{-2}$: (1) 0, (2) $1 \times 10^{12}$, (3) $3 \times 10^{12}$, (4) $7 \times 10^{12}$, (5) $1 \times 10^{13}$, (6) $3 \times 10^{13}$. Dependence of the relative intensity of the 200 XRD peak on the fluence in the linear (**b**) and logarithmic (**c**) axes. Symbol "G" marks the reflection from graphite.

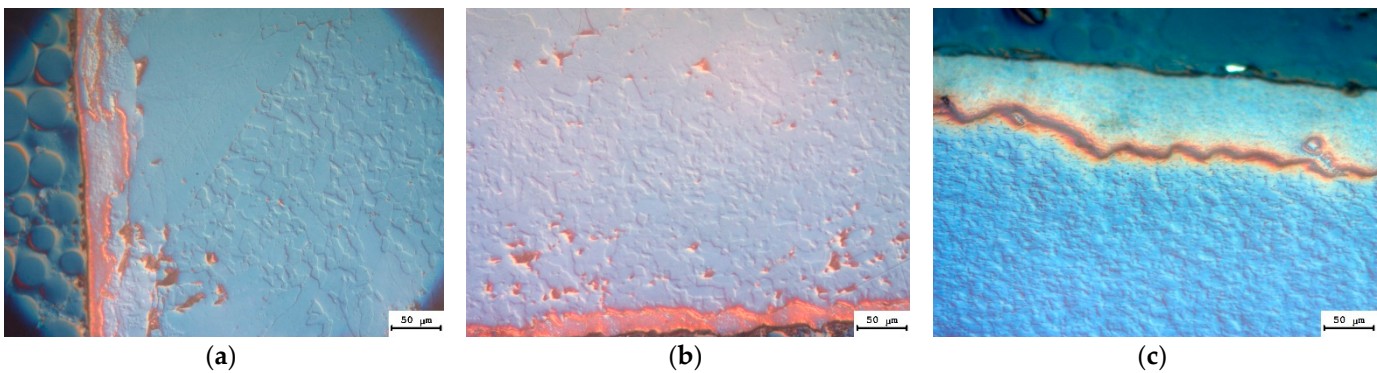

**Figure 7.** Side surfaces of the $YPO_4$:Gd ceramic samples irradiated with different fluences. F, $cm^{-2}$: (**a**) $1 \times 10^{12}$, (**b**) $3 \times 10^{12}$, (**c**) $3 \times 10^{13}$.

The recovery of the sample metamict phase after irradiation with a maximum fluence of $3 \times 10^{13}$ cm$^{-2}$ was studied using sequential annealing with the temperatures increased stepwise from 200 to 700 °C for 3 h at each temperature. The XRD curves were measured after each step (Figure 8). It follows from these data that the crystalline phase had already recovered after annealing at 500 °C (total annealing time was 15 h). Further annealing at 600 °C promoted an increase in the intensity of the reflections. After annealing at 700 °C (total annealing time 18 h), the intensity of the diffraction peaks of the recovered sample reached ~80% of the initial one, $I_0$.

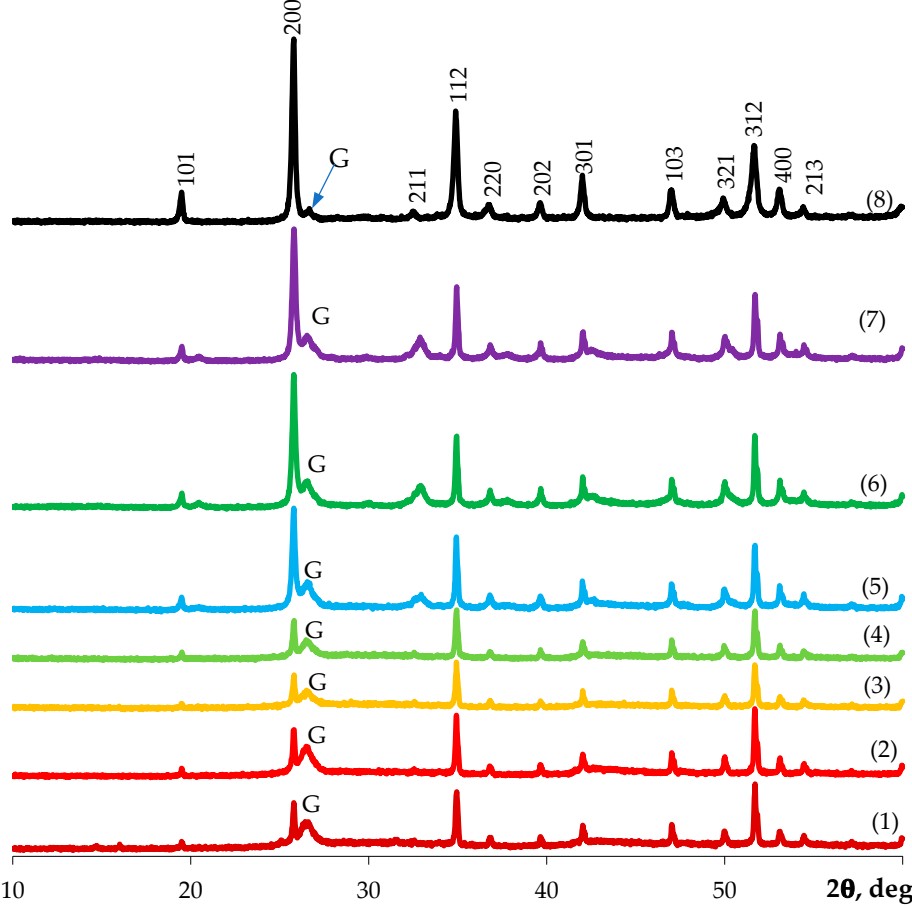

**Figure 8.** XRD data for the ceramic after irradiation (1) and after annealing at 200 °C (2), 300 °C (3), 400 °C (4), 500 °C (5), 600 °C (6), and 700 °C (7). (8) XRD curve from the nonirradiated ceramic sample.

The ceramics obtained were radiation-resistant under the conditions studied in the present work; i.e., they can withstand irradiation from ion beams with high fluences without complete amorphization and can be restored to a highly crystalline state by annealing them.

To analyze the crystal structure of the surface layers of the ceramics in detail, a GIXRD analysis of the initial and irradiated samples was performed. In a series of GIXRD experiments, 200 reflections of the $Y_{0.95}Gd_{0.05}PO_4$ phase were scanned for each sample. The incidence angle of the primary beam on the sample was scanned in a range from 2° to 10° with a step of 1°, which corresponded to variation in the X-ray penetration depth inside the sample from 0.3 to 1.5 μm. The X-ray penetration depth was calculated according to the data from [56]. The calculated X-ray density of the material ($\rho_{th}$) was used in the calculations. It should be noted that the X-ray penetration depth in GIXRD analysis is relatively small (1.5 μm or less at an incidence angle of 10°) and appears to be smaller than the carbonized layer depth estimated from the metallographic investigations. Typical results of the GIXRD analysis for the initial sample and for the ones irradiated with Xe$^{+26}$

ions with a fluence of $7 \times 10^{12}$ cm$^{-2}$ are presented in Figure 9. The same experiments were carried out for all samples in the series.

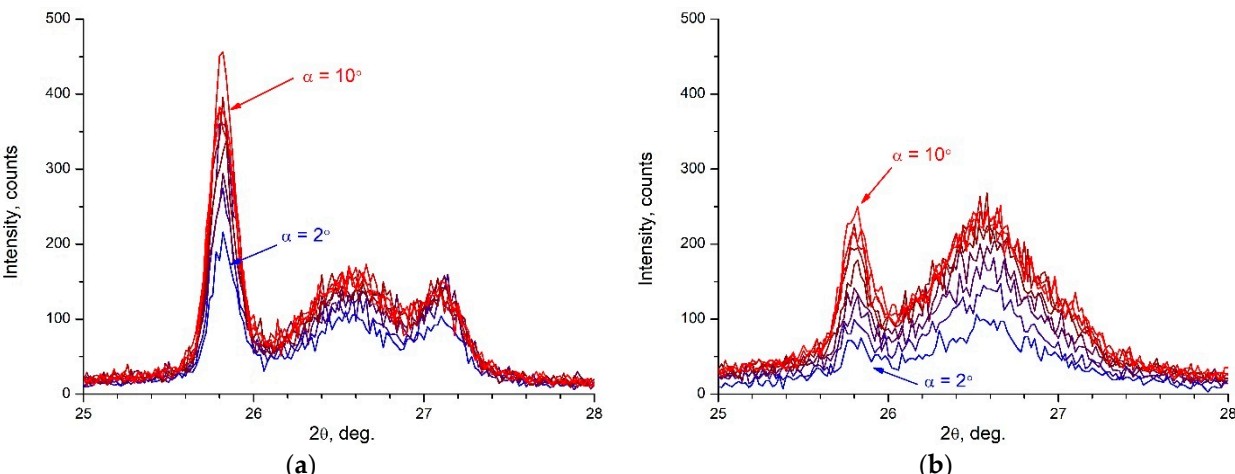

**Figure 9.** GIXRD analysis of 200 reflections of the YPO$_4$:Gd ceramic samples: (**a**) initial state; (**b**) after irradiation with Xe$^{+26}$ ions with a fluence of $7 \times 10^{12}$ cm$^{-2}$.

From the results presented in Figure 9, one can see that the presence of the main YPO$_4$:Gd phase, as well as the graphite one, in the surface layer. In the initial sample, the diffraction peak from the graphite phase was split, which may evidence a nonuniform distribution of this phase inside the sample. In the irradiated sample, the splitting of the graphite peak probably disappeared due to the recrystallization of the graphite phase in the equilibrium conditions after ion irradiation.

To analyze the data from the GIXRD experiments, the dependencies of the integral intensity of the diffraction peaks of the YPO$_4$:Gd phase and the graphite phase on the X-ray incidence angle were plotted. In Figure 10a, these dependencies are plotted (1) for the initial YPO$_4$:Gd sample, (2) for the sample after irradiation with the Xe$^{+26}$ ions with a fluence of $7 \times 10^{12}$ cm$^{-2}$, and (3) for the sample after irradiation with a fluence of $3 \times 10^{13}$ cm$^{-2}$ followed by recovery annealing. The triangles in the figures mark the dependencies of the intensity for the YPO$_4$:Gd phase, and the hexagons mark them for the graphite phase. In addition, Figure 10a shows the additional scale with respect to the recalculation of the incidence angle on the information depth of the XRD analysis. One can clearly see the intensity of the YPO$_4$:Gd phase XRD peak decreases in the irradiated sample down to 30% (at a dose of $7 \times 10^{12}$ cm$^{-2}$) and recovers after annealing back to ~80% of the initial intensity. This evidences the formation of the amorphized surface layer, which partly restores its crystallinity after annealing. The intensity of the graphite phase peak decreased after annealing, which can be explained by the burnout of graphite during annealing.

To analyze the depth distributions of the phases, the dependencies in Figure 10a were normalized in their intensities to an interval of (0;1). The dependencies of the normalized intensities for the diffraction peaks from the YPO$_4$:Gd phase (triangles) and the graphite phase (hexagons) on the primary beam incidence angle on the sample are presented in Figure 10b. One can see the intensity of the diffraction peak from the graphite phase increases with an increasing emission incidence angle on the sample faster than the one from the Y$_{0.95}$Gd$_{0.05}$PO$_4$ phase, both for the initial and irradiated ceramics. This evidences the graphite phase in these samples to be mainly concentrated in the subsurface layer. The irradiation resulted in the amorphization of the YPO$_4$:Gd phase in the whole range of the analysis depths, but it almost did not affect the amorphization of the graphite phase. Annealing resulted in a similar increase in the intensities, for both the main YPO$_4$:Gd phase and the graphite phase, due to an increasing X-ray penetration depth. This evidences a more uniform depth distribution of graphite in the subsurface layer, which can be explained by a burnout of carbon on the surface as well as by its diffusion inside the sample.

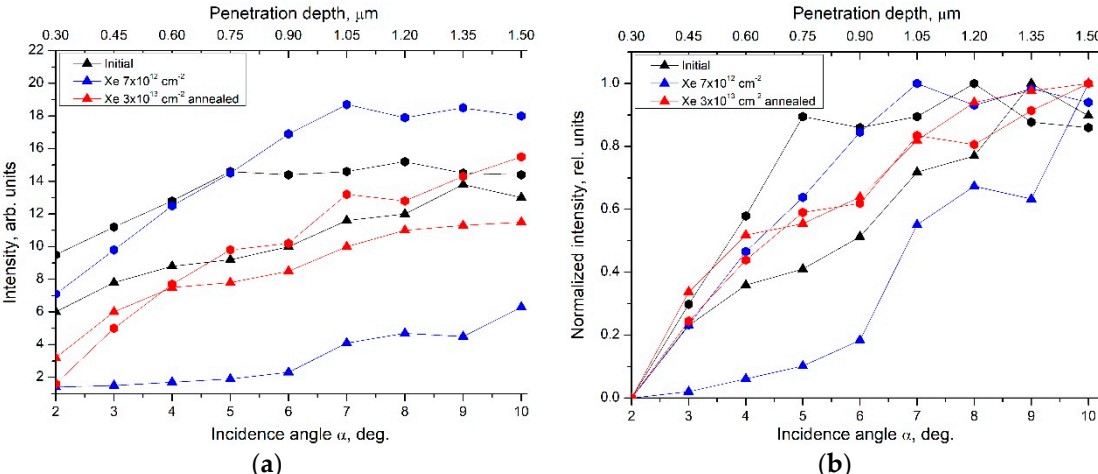

**Figure 10.** Analysis of the GIXRD results: (**a**) dependence of the intensity of the XRD peaks from the YPO$_4$:Gd phase (triangles) and the graphite phase (circles) on the primary beam incidence angle on the sample; (**b**) the same normalized in intensity to the interval (0;1). Black markers—initial sample, blue markers—after irradiation (F = 7 × 10$^{12}$ cm$^{-2}$), red markers—after irradiation and annealing at 700 °C.

The calculations show the amorphization degree of the YPO$_4$:Gd phase in the subsurface layers grow from 20% up to 70% with an increasing fluence, from 10$^{12}$ up to 3 × 10$^{13}$ cm$^{-2}$. After annealing, the sample with the maximum accumulated fluence recovered to a level of 20%.

Finally, let us determine the SPS activation energy for the YPO$_4$:Gd ceramics. To do so, let us use Yang–Cutler model describing the non-isothermic sintering of spherical particles at simultaneous diffusion inside the crystal lattice, grain boundary diffusion, and viscous flow of the material (creep) [57]. The applicability of the Yang–Cutler model to the SPS of ceramics was proved in [35–37,40–42]. In [35–37,40,58–60], this model was applied in order to analyze the high-speed sintering of fine-grained, mineral-like ceramics for HLW immobilization.

According to [57], the slope of the temperature dependence of the relative shrinkage ($\varepsilon$) in the ln(T·$\partial\varepsilon/\partial$T)-T$_m$/T axes corresponds to the effective sintering activation energy, $mQ_{s2}$, where $m$ is a coefficient depending on the dominating sintering mechanism ($m = 1/3$ for the grain boundary diffusion, $m = 1/2$ for the volume (lattice) diffusion, and $m = 1$ for the viscous flow of the material (creep)). In the analysis of the results, the melting point, T$_m$, was accepted to be 2273 K [12].

The dependencies, ln(T·$\partial\varepsilon/\partial$T)-T$_m$/T, had the usual three-stage character (see [57]). At the intensive compaction stage, the dependence ln(T·$\partial\varepsilon/\partial$T)-T$_m$/T can be fitted by a straight line with a good precision (Figure 11). As one can see in Figure 11, the effective activation energy, $mQ_{s2}$, for the YPO$_4$:Gd ceramics was ~7.3 kT$_m$. For $m = 1/3$, typical of SPS in fine-grained ceramics [35–37,40,58–60], the sintering activation energy, $Q_{s2}$, for the YPO$_4$:Gd ceramics was ~22 kT$_m$ (~210 kJ/mol). Unfortunately, there is no data on the activation energy of the grain boundary diffusion in YPO$_4$:Gd ceramics at present. At the same time, it is worth noting that this value of $Q_{s2}$ is close to the activation energy of the grain boundary diffusion for many ceramic materials [35–37,40–42,44,58–60]. This suggests that the accelerated sintering of YPO$_4$:Gd powders in SPS originates from the intensive grain boundary diffusion at ~1200–1400 °C. The grain growth observed in the ceramics (Figure 5), the rate of which also depends on the intensity of the grain boundary diffusion, indirectly supports this suggestion.

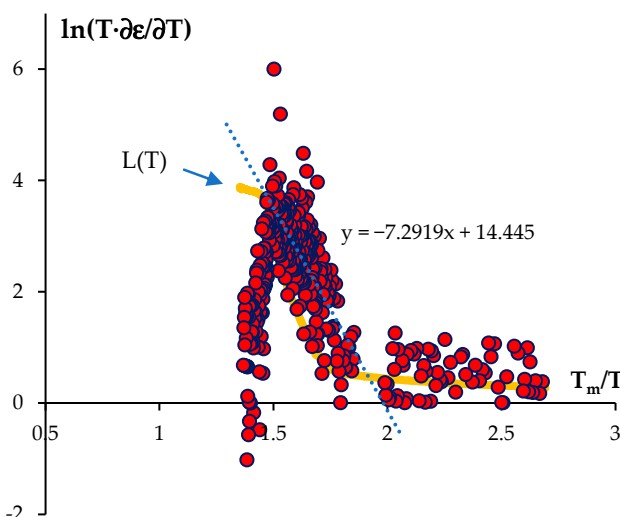

**Figure 11.** The dependence $\ln(T \cdot \partial \varepsilon / \partial T)$-$T_m/T$ for YPO$_4$:Gd ceramic.

## 4. Conclusions

1. The phosphate powders Y$_{0.95}$Gd$_{0.05}$PO$_4$, where Gd acted as a simulator of Cm, were obtained with the sol-gel method. Ceramic samples with a high relative density (>98%) were obtained with spark plasma sintering (SPS). The sintering temperature was 1400 °C, and the whole sintering process duration was ~18 min. The SPS activation energy for the YPO$_4$:Gd fine-grained ceramic was ~22 kT$_m$ (~210 kJ/mol).

2. The ceramic samples demonstrated a high resistance to irradiation from Xe ions with an energy of 167 MeV. At a maximum irradiation fluence of $3 \times 10^{13}$ cm$^{-2}$, the surface layers of the ceramic samples partially retained crystallinity. The calculated value of the fluence leading to the complete amorphization of the surface layers was $(9.2 \pm 0.1) \times 10^{14}$ cm$^{-2}$. After annealing at 500 °C, the metamict phase recovered. After heating at 700 °C, the recovery degree reached ~80%.

3. GIXRD experiments revealed the presence of a graphite phase concentrated mainly near the surfaces of the ceramic samples. Irradiation with high energy ions resulted in the amorphization of the YPO$_4$:Gd phase in the subsurface layers and weakly affected the crystallinity of the graphite phase. Increasing the irradiation fluence resulted in an increase in the amorphization degree of YPO$_4$:Gd from 20% up to 70%. Subsequent annealing of the samples resulted in a decrease in the amorphization degree, down to the level of 20%, as well as in probable burnout and the diffusion of carbon inside the samples, which manifested as a more uniform depth distribution in the graphite phase of the annealed sample.

**Author Contributions:** Conceptualization, A.I.O., D.A.M. and E.A.P.; methodology, D.A.M., E.A.P. and A.I.O.; formal analysis, D.A.M., E.A.P., P.A.Y., V.N.C. and A.I.O.; investigation, D.A.M., E.A.P., P.A.Y., N.V.S., M.S.B., O.A.B., V.A.S., A.T.I. and N.Y.T.; resources, A.I.O., V.N.C. and A.V.N.; data curation, A.V.N., A.I.O. and V.N.C.; writing—original draft preparation, D.A.M., E.A.P. and P.A.Y.; writing—review and editing, A.V.N. and A.I.O.; visualization, A.V.N.; supervision, A.I.O.; project administration, A.V.N.; funding acquisition, A.V.N. All authors have read and agreed to the published version of the manuscript.

**Funding:** This study was funded by RFBR and ROSATOM (Grant No. 20-21-00145). XRD investigations on the specimens after ion irradiation were carried out at the Laboratory of Diagnostics of Radiation Defects in Solid State Nanostructures at the Institute for Physics of Microstructures RAS (IPM RAS) with the financial support of the Ministry of Science and Higher Education of the Russian Federation (Grant No. 0030-2021-0030). The TEM study on the powders was carried out on the equipment of the Center Collective Use "Materials Science and Metallurgy" (National University of Science and Technology "MISIS") with the financial support of the Ministry of Science and Higher Education of the Russian Federation (Grant No. 075-15-2021-696).

**Institutional Review Board Statement:** Not applicable.

**Informed Consent Statement:** Not applicable.

**Data Availability Statement:** Not applicable.

**Conflicts of Interest:** The authors declare no conflict of interest.

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
