# Peer review of "Investigation of the Microstructure of Fine-Grained YPO4:Gd Ceramics with Xenotime Structure after Xe Irradiation"

_ceramics, doi:10.3390/ceramics5020019_

Round 1

Reviewer 1 Report

The paper is devoted to the production of ceramics with xenotime structure by the method of spark plasma sintering (SPS). Y0.95Gd0.05PO4 (YPO4:Gd) phosphates were obtained by sol-gel method. Ceramics had a fine-grained structure and high relative density (98.67%). A comprehensive physical and chemical evaluation of the ceramics obtained, as well as the radiation resistance after Xe irradiation was carried out.

The work is ready to be accepted for printing in its current form.

As an addition it would be interesting to add leaching rates and estimates of diffusion as the ceramic behaves after radiation irradiation 

Author Response

Comment 1. The paper is devoted to the production of ceramics with xenotime structure by the method of spark plasma sintering (SPS). Y0.95Gd0.05PO4 (YPO4:Gd) phosphates were obtained by sol-gel method. Ceramics had a fine-grained structure and high relative density (98.67%). A comprehensive physical and chemical evaluation of the ceramics obtained, as well as the radiation resistance after Xe irradiation was carried out.

The work is ready to be accepted for printing in its current form.

As an addition it would be interesting to add leaching rates and estimates of diffusion as the ceramic behaves after radiation irradiation 

Comment response: The authors thank the referee for the time devoted to our work. We have planned such tests - the tests will be carried out in both stationary mode and dynamic one at several temperatures, including long-term high-temperature autoclave tests. We plan to present the test results later in a separate publication.

Reviewer 2 Report

Dear Authors,

“Investigation of the microstructure of the fine-grained YPO4:Gd ceramics with xenotime structure after Xe irradiation” follows the MDPI's template.

Figure 1. For the b) is missing the x axis.

The article is well written and explained, logical and understandable.

It is also generous in experimental details for and for results

I am puzzled that this text can be found as published, same title same content.

https://ui.adsabs.harvard.edu/abs/2022arXiv220314022M/abstract

The authors should to explain/clarify this coincidence.

I give it a “major revision “because the clarification is important, in the sense that it has not been published before.

Author Response

Comment 1. Figure 1. For the b) is missing the x axis.

Comment response: In Fig. 1b, the x axis, along which the change in the heating temperature [T, oC]) is plotted, begins at the point corresponding to L = 0. This is done for the convenience of analyzing changes in the shrinkage rate of powders during heating.

Comment 2. I am puzzled that this text can be found as published, same title same content. https://ui.adsabs.harvard.edu/abs/2022arXiv220314022M/abstract

The authors should to explain/clarify this coincidence. I give it a “major revision “because the clarification is important, in the sense that it has not been published before.

Comment response: The preprint of this paper was posted on March 26, 2022 on arxiv.org (https://arxiv.org/abs/2203.14022). We noted this in Cover letter. how to send an article to the editorial office, how to send an article to the editorial office of Ceramics. Similar opportunities are provided by MPI publishing house using the platform https://www.preprints.org. The final version of the paper is published under the CC 4.0 license (https://creativecommons.org/licenses/by/4.0/).

In our opinion, this is a very good practice, which allows quick spreading the results obtained among a wide research community as well as receiving comments and suggestions from them to improve the work done.

Reviewer 3 Report

The manuscripts reports a study about Y-xenotime compound doped with gadolinium. The work is well conducted with several data and related interesting discussion. In my opinion, before publications authors should clarify these minor points:

-         - What is the effect of Gd doping in the reported results? In the absence of Gd would the results be different? And Why?

-          - Why authors used a rather complex synthesis procedure to prepare the precursors instead of using solid state procedure, considering the use of a advanced sintering process?

-  Is the achieved very high level of densification necessary for the application?

Author Response

Comment 1. What is the effect of Gd doping in the reported results? In the absence of Gd would the results be different? And Why?.

Comment response: The presence of large Gd3+ cation with larger electron shells than the ones of Y3+ as in the simulated Cm3+ will lead to an increase in the bond length in XO8 polyhedra. This can affect the diffusion properties of ceramics and, for example, the activation energy of sintering. In addition, the addition of Gd3+ cation affects the radiation resistance of the ceramics. The addition of Gd to the YPO4 compound changes the braking capacity of the substance with respect to the xenon ions since it depends on the braking capacity of individual ions in the compound and on the density of the substance itself (the density of Y0.95Gd0.05PO4 is 4.349 g/cm3 whereas the one of in YPO4 is 4.26 g/cm3). As for YPO4, a Gd charged ion spends more of its energy when traveling a shorter path, and therefore such a compound is more radiation-proof. Thus, YPO4 ceramics with Gd will resist to the radiation more effectively when implanted with Xe ions and have smaller depth of the amorphized layer. We also assume that the increased diffusion coefficient in the YPO4:Gd ceramics may contribute to faster recovery of the crystalline phase during annealing, but this assumption needs verification.

Comment 2. Why authors used a rather complex synthesis procedure to prepare the precursors instead of using solid state procedure, considering the use of a advanced sintering process?

Comment response: The choice of the synthesis from solution was motivated by several factors. First, synthesis from solution allows obtaining more uniform distribution of reagents relative to each other during synthesis at lower energy and time costs. Also, the sol-gel synthesis allows obtaining more uniform particle size distribution that is an important condition for obtaining dense ceramics. Second, in the radiochemical industries, it is desirable to eliminate the stages of pouring/grinding if possible and reduce the amount of instrumentation used. Synthesis from solution allows performing all stages of synthesis in one reaction vessel and excluding some stages (for example, grinding).

Comment 3. Is the achieved very high level of densification necessary for the application?

Comment response: High relative density of the ceramics allows increasing its radiation resistance, hydrolytic resistance, and resistance to the thermal shocks. At the moment, there is still no unambiguous dependence of these characteristics on the density – this work has not been completed yet. The results will be published later in separate article.

Round 2

Reviewer 2 Report

The authors' response did not convince me to change my point of view.

I have no other objections to the current version but

I must say that I do not agree with their opinion, that the article is published but still can be classified as "new and unpublished" work.

I keep reading at https://arxiv.org/help/submit, and it doesn't convince me.

Also, I look again to https://arxiv.org/ftp/arxiv/papers/2203/2203.14022.pdf.

(https://arxiv.org/abs/2203.14022)

In my opinion, it is the same thing published in full twice and does not meet the conditions for publication with MDPI.

In fact, in my opinion, it is the same thing published twice, which is not acceptable.

That's why I leave it again to the “major review” and the editorial board to decide on this disagreement.